# Design and Analysis of XY Large Travel Micro Stage Based on Secondary Symmetric Lever Amplification

**DOI:** 10.3390/mi14091805

**Published:** 2023-09-21

**Authors:** Tao Zhang, Liuguang Xiong, Zequan Pan, Chunhua Zhang, Wen Qu, Yuhang Wang, Chunmei Yang

**Affiliations:** 1College of Electromechanical Engineering, Northeast Forestry University, Harbin 150006, China; sxh_00664@163.com (T.Z.); xiongliuguang328@163.com (L.X.); m13736023141@163.com (Z.P.); m17763221024@163.com (C.Z.); quwen314@163.com (W.Q.); 2Forestry and Woodworking Machinery Engineering Technology Center, Northeast Forestry University, Harbin 150006, China

**Keywords:** micromotion stage, PEAs, flexible structure, decoupled kinematics

## Abstract

This study presents a newly developed piezoelectric drive mechanism for the purpose of designing, analyzing, and testing a micro-positioning platform driven by piezoelectric actuators. The platform incorporates a piezoelectric ceramic actuator and a flexible hinge drive and features a symmetrical two-stage lever (STSL) amplification mechanism and a parallelogram output structure. The implementation of this design has led to notable enhancements in the dynamic properties of the platform, thereby eliminating the undesired parasitic displacement of the mechanism. An analytical model describing the fully elastic deformation of the platform is established, which is further verified by finite element simulation. Finally, the static and dynamic performances of the platform are comprehensively evaluated through experiments. A closed-loop control strategy is adopted to eliminate the nonlinear hysteresis phenomenon of the piezoceramic actuator (PEA). The experimental results show that the piezoelectric micro-actuator platform has a motion range of 97.84 μm × 98.03 μm; the output coupling displacement error is less than 1%; the resolutions of the two axes are 8.1 nm and 8 nm, respectively; and the *x*-axis and *y*-axis trajectory tracking errors are both 0.6%. The piezoelectric micromotion platform has good dynamic properties, precision, and stability. The design has a wide application potential in the field of micro-positioning.

## 1. Introduction

The micromotion platform is the essential element of precision testing and manufacturing equipment [1]. It is extensively utilized in the domains of aerospace, optics, ultra-precise machining, and numerous other scientific and technological research fields [2,3,4]. It primarily relies on the elastic deformation of the flexible hinge material to transfer force and displacement; in terms of structure, the flexible hinge has no friction [5], zero clearance [6,7], and high motion sensitivity [8], which results in incomparable advantages when achieving high-precision movement compared to the conventional mechanical transmission [9]. Piezoelectric drive technology [10] is a precision drive technology with considerable advantages such as small size, great accuracy, and no electromagnetic interference. Piezoelectric drive technology is classified into four types: direct-drive piezoelectric drive [11,12,13], ultrasonic piezoelectric drive [14], inchworm piezoelectric drive [15], and stick–slip piezoelectric drive [16]. To obtain a larger motion stroke, it is necessary to amplify the output displacement of the PEA while ensuring high resolution [17,18,19].

Given the wider application of flexible structure micromotion platforms, many scholars at home and abroad have carried out corresponding research. Wu et al. proposed a novel bridge-type compliant displacement amplifier that enhances lateral stiffness and natural frequency in the working direction [20]. They also introduced double parallelogram guide mechanisms at the output end to minimize parasitic movements of the platform. Tian et al. introduced a compact three-degrees-of-freedom micro-nanopositioning platform with an L-shaped lever half-bridge displacement amplifier. This design achieved high amplification ratios for both the *x*- and *y*-axis by utilizing four amplifiers arranged in an orthogonal configuration, resulting in larger amplification ratios and reduced cross-coupling [21]. Gao et al. proposed a flexible XY-bending hybrid piezoelectric-driven platform with characteristics such as low cross-coupling and high resolution [22]. Zhu et al. presented a novel parallel flexible mechanism that combined bridge-type and two-stage lever mechanisms as amplifiers, achieving significant amplification ratios and high-precision trajectory tracking. However, improvements are still needed in terms of natural frequency and resolution. Yuan et al. developed a piezoelectric micro-positioning platform with three-stage amplification and an L-shaped guide, enhancing the platform’s positioning and trajectory tracking capabilities under closed-loop control [23]. However, there is room for optimization regarding displacement coupling errors and motion range. While the aforementioned studies have made impactful contributions in the areas of displacement amplification and reduction in displacement coupling, they have not fully addressed the simultaneous requirements of high amplification ratios and low cross-coupling.

The micro-displacement amplification mechanism mainly includes lever amplification [24], a bridge amplification mechanism [25], a Scott–Russell amplification mechanism, and so on [26,27,28]. The lever mechanism has a flexible structure and high amplification, but the coupling motion is serious, making the actual displacement amplification smaller than the theoretical displacement amplification [29,30,31]. The bridge mechanism is compact, has good symmetry and often poor lateral stiffness, and has difficulty resisting external forces in the direction of non-motion [32,33]; the Scott–Russell mechanism is a triangular amplification mechanism, and, due to the lack of symmetry of the mechanism, and the structure of the bloated, low amplification, it is less used in practical engineering. Therefore, in this paper, a symmetric lever mechanism is used, which not only enhances its output displacement significantly, but also decouples the displacement in the non-motion direction [34].

Based on the concept of a piezoelectric ceramic actuator, (STSL) amplifying mechanism, and extensible hinge, this study proposes the design of a parallel two-dimensional micro-displacement platform. This platform is designed to move with minimal parasitic displacement in the desired direction of motion. The remainder of the paper is structured as follows. Section 2 describes in detail the structure and operating principle of the micro-displacement stage with two degrees of freedom. Section 3 presents a theoretical analysis based on static modeling. In the fourth part, the simulation test is conducted using a workbench to compare the theoretical and simulation results. In the fifth part, a prototype of the platform is built, and the output performance of the platform is tested to provide a comprehensive assessment of the actual performance. The main conclusions are then drawn in Section 6.

## 2. Stiffness Analysis of a Two-Degrees-of-Freedom Micro-Displacement Stage

### Structural Design of a Two-Degrees-of-Freedom Micro-Displacement Stage

This two-degrees-of-freedom micro-displacement stage is based on an STSL amplification mechanism and is designed to satisfy the requirements of a broad working range, a simple structure, and a high intrinsic frequency. The common flexible amplification mechanism can be represented by an array of parallel flexible chains and a series of flexible chains. Two or more flexible chain series are combined to produce a parallel flexible chain. Under the assumption of small deformation, the displacement of the parallel flexible chains is equal to the linear superposition of the displacements of the associated flexible chains under the individual action of each external force and shares a common moving platform to accomplish motion in each direction. The flexible mechanism is essentially a flexible structure composed of a series of rigid and flexible units. The common multi-degrees-of-freedom flexures can be simplified as series and parallel flexures, where the series flexure is made up of multiple single-degree-of-freedom tables superimposed on one another, to the detriment of accuracy and dynamic characteristics, and the displacements of the parallel flexures are equal to the linear superposition of the displacements of the relevant flexures under the separate action of each external force.

Figure 1a depicts the schematic diagram of the two-degrees-of-freedom micro-displacement stage. A table, a frame, and a displacement amplification mechanism compose the platform. Between the input of the displacement amplification mechanism and the frame, the piezoelectric ceramic actuator serves as a displacement output element. The two-degrees-of-freedom micro-displacement stage operates in the *y*-direction. For example, by applying a positive voltage to the PEA after the preload is completed, the PEA transmits the output force to the displacement amplification mechanism. This results in the top of the first-stage lever amplifying mechanism outputting displacement downward in the *y*-direction under the deflection of the straight-circle-type flexible hinge, and the first-stage lever amplifying mechanism of the displacement amplifying mechanism transmits the amplified displacement to the second-stage lever amplifying mechanism, which transmits the displacement to the table using the parallel linkage mechanism. In this case, the L-shaped displacement mechanism can guide the action in the *y*-direction and reduce coupling error in the *x*-direction after amplification. The two-degrees-of-freedom micro-displacement stage realizes symmetrical secondary lever displacement amplification, which reduces output displacement coupling and coupling error by the parallel linkage mechanism, enabling the stage to move in a two-dimensional plane.

The selection of the L-type flexure beam for analysis is based on its exact symmetry and the equal levels of distortion seen in both L-type flexure beams. The table’s deformation is represented in Figure 1b and results in the table’s displacement downward, while the guiding mechanism further strengthens the decoupling capability. Using as an example a two-dimensional piezoelectric micromotor platform operating in the *y*-direction, Figure 1c illustrates the deformation imparted by the force *F* generated by the piezoelectric ceramic actuator for the L-type flexible beam-guiding mechanism with a thickness of *T*.

The size of the PEA’s stroke is comparatively small, necessitating the use of an amplification mechanism to increase the output displacement of the PEA in order to satisfy the stroke requirement. The lever amplification mechanism and symmetrical lever amplification mechanism are two types of amplifiers commonly used in micro-positioning systems, as shown in Figure 2a,b.

The lever amplification mechanism is straightforward in structure, and it is simple to construct multi-stage amplification, but it has a sizeable parasitic displacement. In addition, to achieve a greater amplification ratio, a multi-stage amplification mechanism must be constructed, which will considerably increase the mechanism’s size and make the structure appear obese. Compared with the ordinary lever amplification mechanism, the symmetrical bar amplification mechanism has a larger magnification ratio, and, due to the parasitic deformation of the flexible hinge, the ordinary lever amplification mechanism is prone to parasitic displacement, which has an effect on the positioning accuracy and is not conducive to preventing the piezoelectric ceramic actuator from being affected by the bending moment and shear force. In addition, the STSL amplification mechanism has an output decoupling function to offset the displacement in the non-motion direction.

As depicted in Figure 3, an STSL amplification mechanism based on an oblate flexible hinge type is proposed based on the above analysis. To achieve displacement amplification, a two-stage lever amplification mechanism is intended. Second-stage lever amplification mechanisms are utilized to counteract the parasitic displacement symmetrically. In addition, a parallelogram mechanism at the amplification mechanism’s output reduces output coupling. By applying a positive voltage to the *x*-direction piezoelectric ceramic actuator, as shown in Figure 1, the resulting displacement realizes table movement in the *x*-direction by the lever amplification mechanism and the displacement guide mechanism. As shown in the lower-right corner of Figure 1a, due to the lack of a limiting device, the center axis of the PEA is prone to deviate from the center axis of the input mechanism due to assembly error or vibration, thereby increasing the output coupling. In addition, the PEA may be subjected to a centralized force formed by edge extrusion. By bonding a hemispherical head-type ceramic joint, the concentrated force is evenly distributed to the piezoelectric ceramic plate, preventing the PEA from being damaged.

## 3. Analytical Modeling of Micro-Displacement Platforms

### 3.1. Output Stiffness Analysis

The micro-displacement stage with two degrees of freedom consists of an extensible hinge and a link mechanism. Since the rigidity of the linkage is much higher than that of the flexible hinge, all the motions of the micro-displacement stage are considered to be obtained by the elastic deformation of the flexible hinge, while the linkage is considered to be a rigid body. Flexible hinges are thin elements that provide relative rotation between two adjacent rigid members by bending deformation [35,36,37]. It is employed as a kinematic vice. It is characterized by small size, good stiffness, high sensitivity, and friction. There are two commonly used flexible hinge structures: the straight circular flexible hinge and elliptical flexible hinge [38]. In this paper, we choose the straight circular flexible hinge, whose shape is shown in Figure 4. The dimensional parameters of the micro-displacement platform are shown in Table 1, where the flexible hinge has the minimum thickness, and R is the radius of the flexible hinge.

The height and thickness of the microelement body can be expressed as, respectively:(1)a=t+2r(1−cosθ),
(2)du=d(sinθ)=rcosθdθ,

As the dimension of the flexible hinge at the incision is much smaller than at other locations, a very small change in bending moment can be identified here, and flexible hinges are not easy to deform. The deflection curve is particularly flat when *t* ≦ 0.2 *R* at the time of a straight circular flexible chain according to the bending deformation of the mechanics of materials whereby it is known that the flexural curve can be obtained by the flexure curve of the flexible chain of the differential equations.
(3)$$\frac{1}{\rho }=\frac{y^{\prime \prime }}{{\left(1+y^{\prime 2}\right)}^{\frac{3}{2}}}=\frac{\frac{d^{2}y}{dx^{2}}}{{\left[1+{\left(\frac{dy}{dx}\right)}^{2}\right]}^{\frac{3}{2}}}=\frac{M\left(x\right)}{EI\left(x\right)},$$
where *E* is the modulus of elasticity, and *M*(*x*) and *I*(*x*) are the bending moment and moment of inertia applied to the microelement *dx*, respectively.

The flexible hinge, by the bending moment in its incision, changes very little. According to the knowledge of higher mathematics, it can be simplified. At the same time, at the flexible hinge, bending moment *M*(*x*) constant processing can be generated; the deformation of the flexible hinge is small around the curve and is flat to meet $\frac{dy}{dx}\ll 1,$. Therefore, it can be simplified as:(4)$$\frac{d^{2}y}{dx^{2}}=\frac{M\left(x\right)}{EI\left(x\right)},$$

Bringing the inertia I(x)=bh(x)/12 of the cross-section of the microelement *dx* against the central axis into the above equation and integrating it yields the formula for the angle of rotation of the flexible hinge *θ*:(5)$$\theta =\frac{dy}{dx}=\int \frac{d^{2}y}{dx^{2}}dx=\int \frac{12M}{Ebh\left(x\right)}dx,$$

To facilitate the calculation, it is first simplified and can be obtained by transforming the right-angle coordinate system into the form of a polar coordinate system with the center of the circle of the flexible hinge as the coordinate origin.
(6)$$\left\{ \begin{array}{l} dx=d\left(R+r\sin\varphi \right)=r\cos\varphi d\varphi \\ h\left(x\right)=t+2r-2r\cos\varphi \end{array}\right.,$$

Substituting the above two equations into Equation (5) gives:(7)$$\theta =\frac{12M}{Eb}\int_{-\frac{\pi }{2}}^{\frac{\pi }{2}}\frac{r\cos\varphi }{\left(t+2r-2r\cos\varphi \right)}d\varphi ,$$

Substituting $s=\frac{r}{t}$ into Equation (7) yields:(8)$$\theta =\frac{12M}{Eb}\int_{-\frac{\pi }{2}}^{\frac{\pi }{2}}\frac{r\cos\varphi }{\left(t+2r-2r\cos\varphi \right)}d\varphi =\frac{12M}{EbR^{2}}\int_{-\frac{\pi }{2}}^{\frac{\pi }{2}}\frac{\cos\varphi }{\frac{1}{s}+2-2\cos\varphi }d\varphi =\frac{12Mf_{1}}{EbR^{2}},$$

The force and torque of the flexure hinge on the *x*-, *y*-, and *z*-axis are $F_{x}$, $F_{y}$, $F_{z}$, $M_{x}$, $M_{y}$, and $M_{z}$. Assuming that the left end of the flexure hinge is fixed, under the action of $F_{y}$, the linear deformation Δy along the *y*-axis is caused by the bending moment of straight circular flexure hinge, and stiffness can be expressed as:(9)$$K_{1}=\frac{F_{y}}{\Delta y}=\frac{Eb}{12\sin^{2}\theta f_{1}+12f_{2}}$$
where the value of $f_{1}$ and $f_{2}$ are coefficients given by:(10)$$\begin{array}{l} f_{1}=\frac{8s^{4}(2s+1)}{{\left(4s+1\right)}^{2}}\frac{\tan\frac{\beta }{2}}{\left[1+(4s+1)\tan^{2}\frac{\beta }{2}\right]}+\frac{4s^{3}(6s^{2}+3s+1)}{{\left(4s+1\right)}^{2}}\frac{\tan\frac{\beta }{2}}{1+(4s+1)\tan^{2}\frac{\beta }{2}}\\ +\frac{12s^{4}(2s+1)}{{\left(4s+1\right)}^{\frac{5}{2}}}\arctan\left(\sqrt{4s+1}\tan\frac{\beta }{2}\right), \end{array}$$
(11)$$\begin{array}{l} f_{2}=-\frac{2s^{4}\left(2s+1\right)}{\left(4s+1\right)}\frac{\tan\frac{\beta }{2}}{{\left[1+(4s+1)\tan^{2}\frac{\beta }{2}\right]}^{2}}+\frac{s\left(2s^{2}+5s+1\right)}{\left(4s+1\right)}\frac{\tan\frac{\beta }{2}}{1+\left(4s+1\right)\tan^{2}\frac{\beta }{2}}\\ +\frac{\left(2s+1\right)\left(2s^{2}-4s-1\right)}{2{\left(4s+1\right)}^{\frac{3}{2}}}\arctan\left(\sqrt{4s+1}\tan\frac{\beta }{2}\right)+\frac{\beta }{4}, \end{array}$$

From the above equation, the rotational stiffness $k_{2}$ of the straight cylindrical flexible hinge around the *z*-axis is:(12)$$k_{2}=\frac{M}{\theta }=\frac{Ebr^{2}}{12f_{1}},$$

According to the series–parallel relationship of stiffness, the input stiffness $K_{D_{in}}$ of the whole displacement amplification mechanism can be obtained.
(13)$$K_{D_{in}}=\frac{2K_{1}K_{2}^{2}}{2K_{1}K_{2}+K_{2}^{2}},$$

### 3.2. Static Analysis

To ensure that the XY micro stage can work within a reasonable elastic deformation working range when the piezoelectric ceramic actuator is generated to reach the maximum output displacement, the maximum stress generated by the micro stage is much lower than the yield stress of the material. For this phase, the maximum stress $\sigma_{max}$ occurs when the maximum rotation angle $\omega_{max}$ of one of the upper-left circular flexible hinges reaches a maximum; the relationship between the bending stress and deformation during the deformation of the circular surface is as follows [39,40]:(14)$$\sigma_{max}=\frac{E{\left(1+\beta \right)}^{\frac{9}{20}}}{\beta^{2}f\left(\beta \right)}\omega_{max}<\frac{\sigma_{y}}{S},$$
(15)$$f\left(\beta \right)=\frac{1}{2\beta +\beta^{2}}\left[\frac{3+4\beta +2\beta^{2}}{\left(1+\beta \right)+\left(2\beta +\beta^{2}\right)}+\frac{6\left(1+\beta \right)}{{\left(2\beta +\beta^{2}\right)}^{\frac{3}{2}}}\tan^{-1}{\left(\frac{2+\beta }{\beta }\right)}^{\frac{1}{2}}\right],$$
where β=t/2R, *t* and *R* are the thickness and radius of the circular flexible hinge, respectively, f(β) is a function of β, $\omega_{max}$ is the rotation angle of the hinge, and *S* is the safety factor.

### 3.3. Dynamic Analysis

In this paper, dynamic modeling is performed using the Lagrange method, which analyzes the dynamic characteristics of the system from the standpoint of energy conservation and avoids the laborious study of minutiae required by the Newtonian method. Considering the dynamic characteristics of the micromanipulator platform, the first-order intrinsic frequency of the micromanipulator platform is investigated in this paper. It should be that the characteristics along the *x*-axis and the *y*-axis are the same, and only the *y*-axis is analyzed dynamically.

In the kinetic analysis, all flexible hinges are reduced to an ideal kinematic pair with kinematic-oriented stiffness. From the kinematic point of view, the input displacement $u_{in}$ is set to generalized coordinates for ease of description, and the kinetic energy of the system can be expressed by the following equation:(16)$$T=\frac{1}{2}\left(\sum_{i=1}^{n}C_{in}^{i}m_{i}\right)u_{in}^{2},$$
(17)$$C_{in}^{i}=\frac{u_{i}}{u_{in}},$$
where $m_{i}(i=1,2,3\ldots 5,6)$ is the mass of the flexible hinge fast, $C_{in}^{i}$ is the conversion factor, and $u_{in}$ is the displacement of the rigid fast.

The potential energy of the system is mainly reflected in the elastic potential energy of the flexible hinge.
(18)$$V=\sum_{i=1}^{n}\frac{1}{2}k_{i}\theta_{i}^{2}=\frac{1}{2}\left(\sum_{i=1}^{n}C_{in}^{i}k_{i}\right)u_{in}^{2},$$

The conversion factor can be obtained by calculating the hydrostatic equation. Substituting kinetic and potential energies into Lagrange’s equation gives: (19)$$\frac{d}{dt}\frac{\delta T}{\delta u_{in}}-\frac{\delta T}{\delta u_{in}}+\frac{\delta V}{\delta u_{in}}=F_{in},$$

Using Equations (16)–(19), the kinetic expression for the micromotion platform can be derived.
(20)$$M_{e}{\ddot{u}}_{in}+K_{e}u_{in}=0,$$
(21)$$\left\{ \begin{array}{l} M_{e}=\sum_{i=1}^{n}c_{in}^{i}m_{i}\\ K_{e}=\sum_{i=1}^{n}c_{in}^{i}k_{i} \end{array}\right.,$$

$M_{e}$ and $K_{e}$ are the equivalent mass and equivalent stiffness in the *x*-axis direction of the micro stage, respectively.

Solving Equation (21), the first-order intrinsic frequency of the micromotion stage is obtained as:(22)$$f=\frac{1}{2\pi }\sqrt{\frac{K_{e}}{M_{e}}}=\frac{1}{2\pi }\sqrt{\frac{\sum_{i=1}^{n}c_{in}^{i}k_{i}}{\sum_{i=1}^{n}c_{in}^{i}m_{i}}},$$

## 4. FEA Validation of Micro-Displacement Platforms

In order to ensure the correctness and reliability of the design, we carried out finite element simulation analysis of the two-degrees-of-freedom micro-displacement stage using finite element analysis software to verify the static and dynamic performance of the platform. The material density, Young’s modulus, Poisson’s ratio, and yield strength of 65 Mn were selected as the material of the micro-displacement stage, which were 7820 kg/m^3^, 210 GPa, 0.282, and 785 MPa, respectively. The micro-displacement platform was simulated using finite element analysis software. The automatic meshing method was used to establish the mesh model initially, and the flexible mechanism was refined manually to obtain more accurate simulation results.

### 4.1. Static Analysis Validation

First, using finite elements, the amplification ratio and output stiffness of the micro-displacement stage were analyzed when a constant 100 N force was applied to the actuator’s input. Figure 5a depicts the deformation of the micro-displacement stage. By applying a constant force to one of the inputs and detecting the corresponding output displacement u of the platform, the input stiffness $k_{in}=F/u_{in}=5.56$ N/μm. In addition, a constant force of 100 N was applied to the *y*-direction of the end effector, and the platform was calculated to have an output stiffness $k_{out}=$ 0.18 N/μm. It was shown that the results of the finite element analysis matched well with those of the analytical calculations.

When one of the inputs has a displacement of 10 μm in the *y*-direction, the amplification ratio $\lambda =u_{y}/u_{in}$ of the platform can be determined by comparing the output displacement $u_{y}$ of the end effector along the *y*-axis direction divided by the input displacement $u_{in}$. The amplification ratio obtained by the finite element analysis was about 5.27, which is basically the same as that of the results of the analytical model (5.31), and the corresponding deviation was only 1.9%. Under the same operating conditions, it is assumed that the parasitic displacement of the output at the other end along the *y*-axis driving direction is $\Delta y_{in}$, and the displacement of the end effector along the transverse direction is $\Delta y_{out}$. The input–output displacement coupling ratios are calculated as $\varepsilon_{in}=\Delta_{in}/u_{in}$ and $\varepsilon_{out}=\Delta y_{in}/u_{y}$, respectively.

In addition, the stress distribution of the micro-displacement stage was simulated at an input displacement of 15 μm on the *x*-axis with a PEA stiffness of 60 N/um and a nominal displacement of 20 μm at maximum voltage, as depicted in Figure 5b. Parasitic displacement in the finite element simulation results showed that when the micro-displacement platform was driven in the vertical direction, the maximum stress value of 93.5 MPa was simulated, which is close to the 98.7 MPa calculated by Equation (14). More crucially, it is significantly less than the yield stress of the chosen material (430 MPa). As a result, the micro-displacement platform will always operate inside the full workspace’s elastic deformation range.

### 4.2. Dynamic Analysis and Validation

The modal analysis of the micro-displacement platform was performed using finite element analysis. The platform’s inherent frequency was determined, and the structural design’s viability was validated. The frame of the micro-displacement platform was fixed, and modal analysis was performed. As shown in Figure 6, the first four orders of the intrinsic frequencies of the micro-displacement platform are 204.5 Hz, 267.1 Hz, 577.5 Hz, and 1121.6 Hz, respectively. The resonance frequency of the simulation result of 204.5 Hz is the same as that of the analytical model result of 206.1 Hz with a difference of 0.77%.

Under the first vibration mode, the deformation of the micromanipulator platform in the *x*- and *y*-direction is the same, which shows the similar dynamic characteristics of the micromanipulator platform in different directions; the second vibration mode shows the deflection coupling of the platform in the *x*- and *y*-direction; and the third vibration mode shows the micromanipulator platform rotating counterclockwise in the plane. The fourth vibration mode is the intrinsic frequency under the undesired mode, which is 78% higher than the first vibration mode. It can be seen that the micromotion platform has good dynamic performance.

As shown in Table 2, the two coupling ratios derived from the analytic model are 0.05 and 0.1 percent. FEA simulated ratios of approximately 0.08% and 0.12%. The error in numerical calculation contributed to the non-zero coupling ratios. The low values of the coupling ratios indicate that coupling effects can be completely disregarded. Therefore, the developed XY micro and nanopositioning platform stage is entirely decoupled in both the *x*- and *y*-direction at both the input and output extremities.

## 5. Experimental Study of Micro-Displacement Platforms

### 5.1. Experimental Setups

In order to validate the theoretical properties and simulation analysis of the micro-displacement platform, a platform structure prototype and a two-degrees-of-freedom micro-displacement platform sample were fabricated using wire-cut machining and 65 Mn. The prototype platform was attached to the optical table with bolts. The theoretical characterization and finite element simulation of the two-degrees-of-freedom micro-displacement stage were verified, as shown in Figure 7a. The dimensions of the two-degrees-of-freedom micro-displacement stage were 208.9 mm × 208.9 mm × 10 mm. Subsequently, a series of experiments were conducted to validate the two-degrees-of-freedom micro-displacement platform prototype. To prevent external interference, the micro-displacement platform was placed on a vibration-isolated optical stage, and both open-loop and closed-loop experiments were conducted; in the open-loop experiments, displacement coupling, stiffness, step response, and frequency response experiments were conducted on the platform to test the actual static–dynamic characteristics of the platform. In the closed-loop experiments, to test the positioning and trajectory tracking ability of the platform, step response and trajectory tracking experiments were carried out. The input displacement was provided by PEAs with two integrated strain gauges (type Pst150/7/20VS12, COREMORROW). The PEAs with built-in integrated strain gauges have a drive voltage of 0–150 V, a nominal displacement of 20 μm, and a stiffness of 60 N/μm. Two laser displacement sensors (SG-5020, SSZN) with a repeatability of 0.02 μm were used, as shown in Figure 7b; the sensors are based on the principle of triangulation of light measurement through the sensor target to measure the output displacement, which was used to collect the displacement data of the micromanipulation platform; the resolution experiments were performed using a confocal sensor to acquire the data, and the data were transmitted to the PC through the sensor controller.

### 5.2. Open-Loop Experiments

The high hysteresis characteristic of the piezoelectric ceramic actuator impacts the speed and precision of positioning when the micro stage is utilized. Consequently, it was imperative to investigate the hysteresis properties of the micro stage. First, the hysteresis curve of the micro-actuator platform was given, as shown in Figure 8. The platform was tested, and the experiment was carried out under open-loop conditions; we ran three sets of tests to gain a better understanding of the effect of hysteresis qualities, and the experimental results are shown in Figure 8a; it can be observed that the input voltage $V_{x}$ = 0 V rises to $V_{x}$ = 100 V and then falls to $V_{x}$ = 0 V. When the input voltage is 100 V, the motion displacement of the micro stage increases to $l_{x}$ = 78.86 μm. For the purpose of error analysis, we repeated the experiment three times with good repeatability. The maximum displacement error of the piezoelectric ceramic actuator during expansion and contraction is $e_{x}$ = 17 μm, and the hysteresis ratio of the amplifying mechanism platform along the *x*-axis is 13.83%. Figure 8b shows the maximum travel of the micro-actuator platform; the input voltage is increased from 0 to 150 V. The maximum output displacement of the micro-actuator platform is $l_{x}$ = 97.13 μm when the input voltage is 150 V. At the same time, the built-in strain gauges of the piezoelectric ceramic actuator can detect the output displacement, the output displacement of the piezoelectric ceramic actuator is $i_{x}$ = 21.24 μm when the input voltage is 150 V, and the micro-actuator amplification ratio ζ of the platform is 4.59.

This study focused on examining the platform’s operational displacement, amplification ratio, and decoupling performance. Initially, a sinusoidal voltage signal with a peak amplitude of 80 V and a frequency of 1 Hz was applied to the piezoelectric actuator (PEA) along the *y*-axis. Subsequently, the micromanipulator platform was displaced in the *y*-axis direction. The output coupling in the *y*-axis direction was detected by measuring the displacement on the *y*-axis and the displacement on the *x*-axis of the output through a sensor. The measured sinusoidal motion of the output, as well as the coupled motion, is shown in Figure 9a,b. The amplification ratio of the stage in the *y*-direction deviated from the theoretical and simulation results, and the possible reasons for the error may be: (1) asymmetry of the stage due to manufacturing and machining errors and (2) different preload conditions of the PEA.

### 5.3. Closed-Loop Experiments

The study aimed to assess the positioning capacity and trajectory tracking of a two-degrees-of-freedom micromanipulator stage by implementing the proportional–integral–derivative (PID) control approach in a closed-loop control system. The adjustment of PID parameters in this study was conducted using a trial-and-error approach. The control gain was meticulously tuned to achieve precise placement. Furthermore, the controller acquired the discrepancy between the reference and actual signals. The sampling frequency of the measured data was 100 kHz. The step response along the *y*-axis is shown in Figure 10a. The target displacement was set to 23.8 μm, and the steady-state displacement deviation and error were ±0.02 μm and 0.6%, respectively. As can be seen from the figure, there was almost no overshooting, and the settling time was about 80 ms. As shown in Figure 10b, the target trajectory was circular, and the platform could accurately track the circular trajectory within the trajectory tracking range of 0–9 μm. Therefore, in practical applications, the closed-loop controlled micromotion platform can realize a fast response and accurate positioning functions.

Finally, to more accurately obtain the motion resolution of the entire micro stage in both directions, a piezoelectric ceramic actuator from COREMORROW (pst150/7/20 VS12) was used with a resolution of 2 nm. The piezoelectric ceramic actuator was manipulated to induce a stepping motion, resulting in a step displacement of 2 nm. The step displacement at the output end was quantified using a spectral confocal displacement transducer, which had a resolution of 6 nm. As shown in Figure 11, the sampling frequency was set to 10 kHz, and the observed jitter was mainly due to the influence of ambient noise on the experiment. Through the multistep response test, shown in Figure 11a,b, it was indicated that the resolvable resolution of the *x*- and *y*-axis of the micromotion platform was 8.1 nm and 8 nm, respectively, which may have been caused by machining errors, errors due to assembly, and noise interference in the environment.

Swept frequency experiments were conducted to validate the dynamic characteristics of the piezoelectric micro-displacement stage. The piezoelectric ceramic actuator was subjected to a sinusoidal excitation signal with a sweeping frequency. The actuator’s voltage amplitude was 80 V, and the voltage was non-negative. The frequency of the signal was raised from 1 Hz to 250 Hz. The frequency response characteristics of the piezoelectric micro-actuator platform could be obtained by performing Fast Fourier Transformation (FFT) on the input voltage and the displacement at the output. As shown in Figure 12a, the first-order intrinsic frequency was 211 Hz along the *x*-axis, and Figure 12b indicates 206 Hz along the *y*-axis. Although the intrinsic frequencies of the *x*- and *y*-axis reversals are theoretically the same, they may differ due to the fabrication and assembly errors of the piezoelectric micro-actuator stage. Nevertheless, the analytical results are in good agreement with the finite element simulation results. The relative deviations between the experimental results and the finite element simulation results for the two modes are 3.3% and 0.97%, respectively.

The frequency response tests were carried out to verify the dynamic characteristics of the piezo-actuated platform. A sinusoidal signal of 75 V with frequencies of 1 Hz, 10 Hz, 80 Hz, and 120 Hz was applied to the piezoelectric ceramic actuator. The laser displacement sensor was employed to measure the displacement responses. Figure 13 depicts the displacement of the emission of sinusoidal signals at various frequencies. When the frequency was 1 Hz and 10 Hz, it can be seen that the displacement deviation of the micro-positioning stage in the *x*-axis direction was relatively stable. When the frequency was 80 Hz, the tracking error performance of the micromanipulation platform was 0.02 μm, and even when the frequency was increased to 120 Hz, the tracking error performance remained at 0.02 μm. This indicates that the platform’s dynamic performance is satisfactory.

## 6. Conclusions

In this study, a novel STSL amplification mechanism is proposed. Furthermore, a piezoelectric micro-positioning platform with two degrees of freedom is evaluated. A piezoelectric ceramic actuator and a flexible hinge operate the platform. In contrast to prior designs, the mechanism presented in this study aims to achieve two-stage displacement amplification and motion direction modulation. In addition to the simple design and symmetrical structure of the traditional lever mechanism, it also has the characteristics of output decoupling. The analytical model describing the full elastic deformation behavior of the platform is established, and the kinematics, maximum stress, and dynamic characteristics of the micro-moving platform are understood and modeled. The results are further verified by finite element simulation.

A prototype of the piezoelectric micro dynamic platform was fabricated, and the static and dynamic performances of the platform were comprehensively evaluated through a series of experiments. The piezoelectric micromotion platform’s output properties were also evaluated. The experimental results showed that the amplification ratios of *x*- and *y*-direction motions were 4.5 and 4.6, respectively; the range of motion was 97.8 μm × 98.03 μm; the output coupling error was less than 1%; the hysteresis rate was 13.83%; the resolutions of the two axes were 8.1 nm and 8 nm, respectively; the first-order intrinsic frequency along the *x*-axis was 211 Hz; and, along the *y*-axis, the first-order intrinsic frequency was 206 Hz. When applying a 120 Hz sinusoidal excitation signal, the *x*-axis and *y*-axis trajectory tracking errors were both 0.6%. Finally, the piezoelectric ceramic actuator’s closed-loop control via strain gauge integration efficiently removed the piezoelectric actuator’s nonlinear hysteresis. The results show that the positioning platform can accurately move along the circular trajectory.

## Figures and Tables

**Figure 1 micromachines-14-01805-f001:**
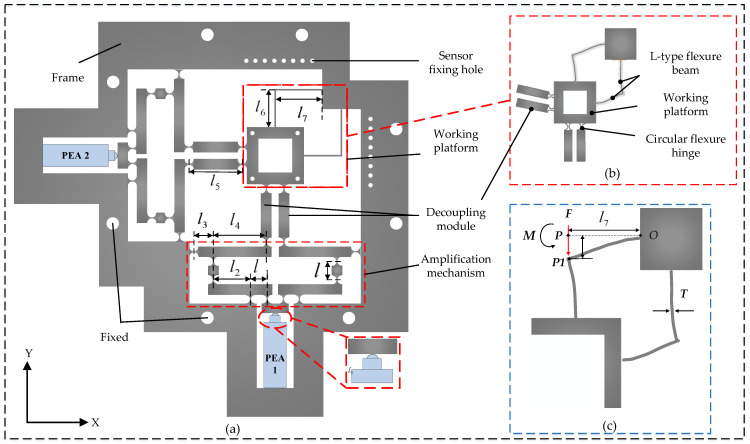
Structure of the 2D piezoelectric platform. (**a**) Two-degrees-of-freedom micro-displacement platform; (**b**) micro table displacement diagram; (**c**) deformation diagram of L-type flexure beam.

**Figure 2 micromachines-14-01805-f002:**
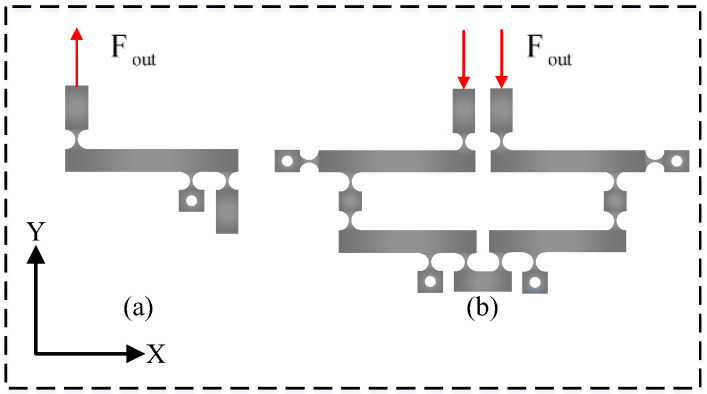
(**a**) Lever amplification mechanism; (**b**) STSL mechanism.

**Figure 3 micromachines-14-01805-f003:**
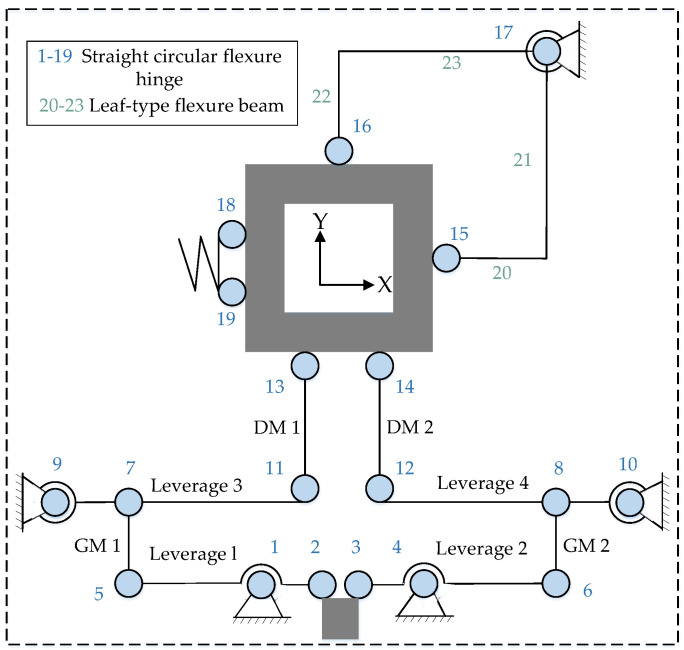
Geometric model of the displacement amplifying mechanism and schematic diagram of the platform analysis model.

**Figure 4 micromachines-14-01805-f004:**
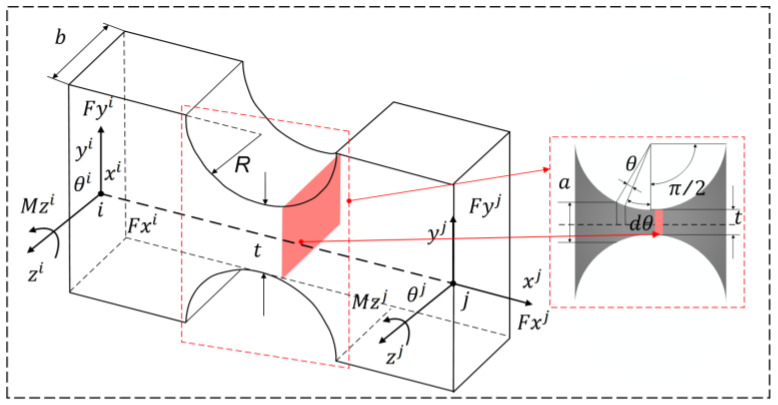
Straight circular flexible hinge and schematic diagram of microelement division.

**Figure 5 micromachines-14-01805-f005:**
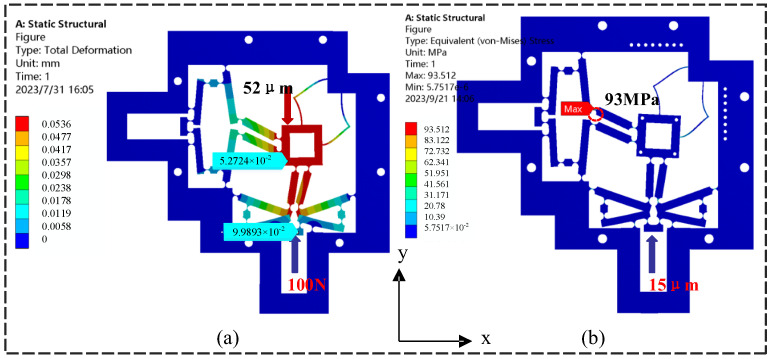
Static analysis results of the finite element method. (**a**) Platform output displacement map, (**b**) convergence of the equivalent stress.

**Figure 6 micromachines-14-01805-f006:**
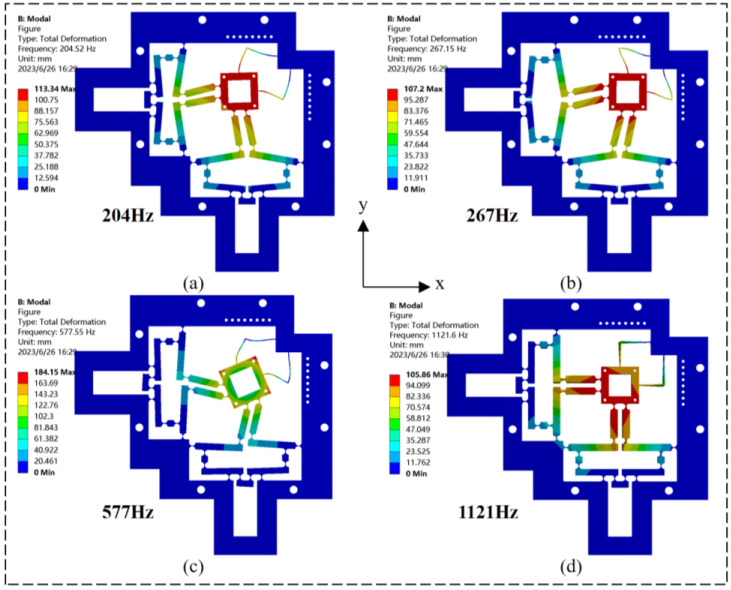
Dynamic analysis results of finite element method. (**a**) First vibration mode. (**b**) Second mode. (**c**) Third mode. (**d**) Fourth mode.

**Figure 7 micromachines-14-01805-f007:**
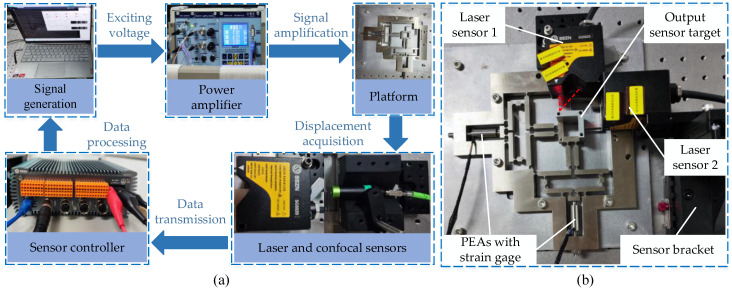
Prototype and experimental setup. (**a**) Experimental process. (**b**) Fretting platform setting.

**Figure 8 micromachines-14-01805-f008:**
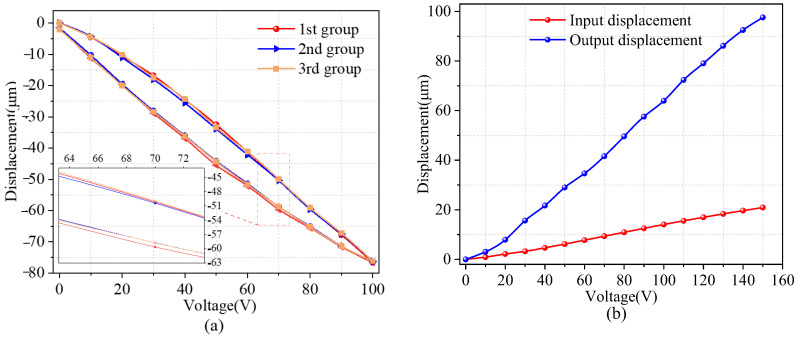
(**a**,**b**) Hysteresis characteristic curves and maximum displacement diagram, respectively.

**Figure 9 micromachines-14-01805-f009:**
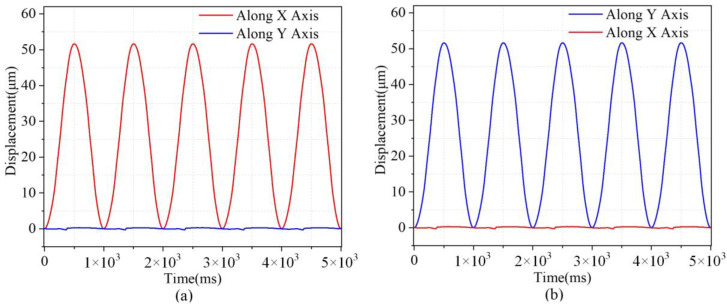
Output displacements and the corresponding cross-couplings of the piezo-actuated platform. (**a**) Excited on the *x*-axis. (**b**) Excited on the *y*−axis.

**Figure 10 micromachines-14-01805-f010:**
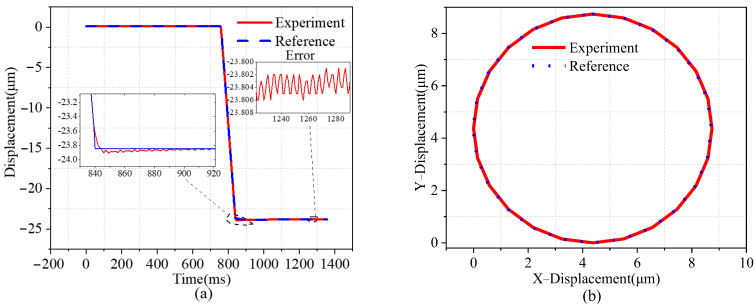
Point-to-point positioning control. (**a**) Experimental results of the step jump response of the 2D piezoelectric platform. (**b**) Experimental results of circular trajectory tracking on the micro-moving platform.

**Figure 11 micromachines-14-01805-f011:**
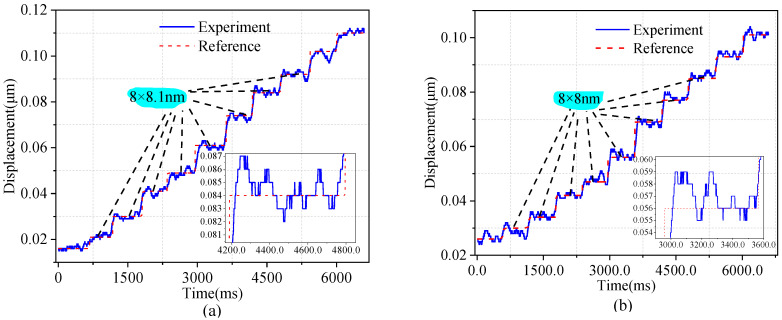
Motion resolution of the micromotion platform (**a**) along the *x*-axis, (**b**) along the *y*−axis.

**Figure 12 micromachines-14-01805-f012:**
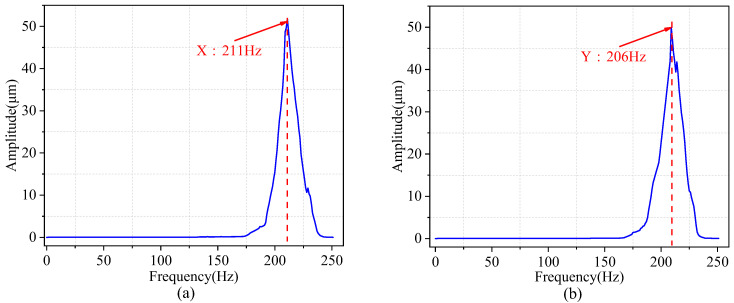
Frequency response of the fretting platform (**a**) along the *x*-axis, (**b**) along the *y*-axis.

**Figure 13 micromachines-14-01805-f013:**
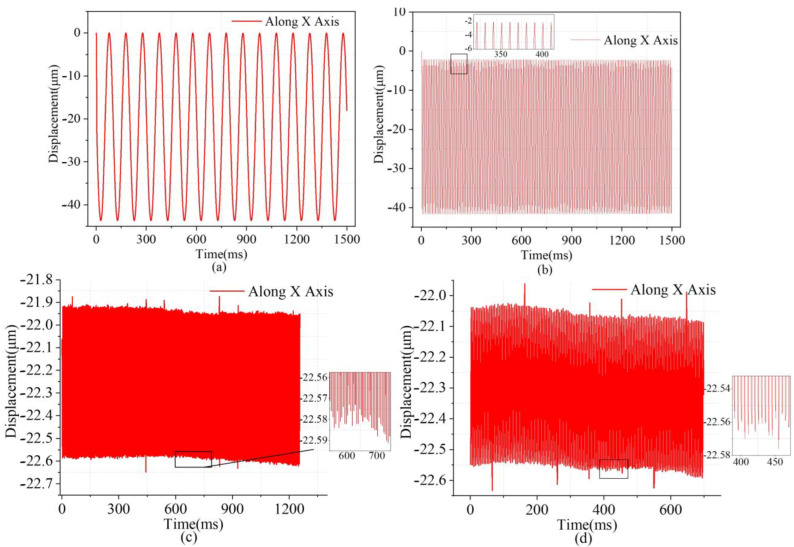
Output displacement curves at different sinusoidal frequencies: (**a**) 1 Hz, (**b**) 10 Hz, (**c**) 80 Hz, (**d**) 120 Hz.

**Table 1 micromachines-14-01805-t001:** Dimensional parameters.

Para	$l_{}$	b	t	R	T	$l_{1}$	$l_{2}$	$l_{3}$	$l_{4}$	$l_{5}$	$l_{6}$	$l_{7}$
Value (unit: mm)	10	10	0.4	2.5	0.8	9	19.6	10.2	28	28	12.3	16.8

**Table 2 micromachines-14-01805-t002:** Performance of the micromotion platform.

	ζ	$k_{in}$	$k_{out}$	$\varepsilon_{in}$	$\varepsilon_{out}$	$\sigma_{max}$	$f_{1}$
Anal.	5.69	6.8 N/μm	0.21 N/μm	0.0005	0.001	98.7 MPa	206.1 Hz
FEA	5.8	5.56 N/μm	0.18 N/μm	0.0008	0.0013	93.5 MPa	204.5 Hz
Error	1.9%	18.1%	14.2%	5%	5.8%	5.3%	0.78%

## Data Availability

The authors confirm that the data supporting the findings of this study are available within the article. In addition, the data that support the findings of this study are available from the corresponding author, Yuhang Wang, upon reasonable request.

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
