# Peer review of "Design and Analysis of XY Large Travel Micro Stage Based on Secondary Symmetric Lever Amplification"

_micromachines, 2023, doi:10.3390/mi14091805_

Round 1
Reviewer 1 Report
In the manuscript, Zhang et al. presented a novel piezoelectric drive mechanism to design, analyze, and test a two-degree-of-freedom piezoelectric-driven micro-positioning platform. According to the experimental results, the output coupling displacement error is less than 1%, and the trajectory tracking errors of the X-axis and Y-axis are both 0.6%. Although this work is interesting, I have some comments before this work is further considered in Micromachines.
1. English language should be revised thoroughly, such as “the X-axis and Y-axis The trajectory tracking error is 0.6% for both X-axis and Y-axis (Rows 444 and 445)” and so on, please carefully check through.
2. The equations have the following problems: (1) the equation number is wrong, (2) the variables in equations should be italicized when they appear in the manuscript, and (3) the font in equations does not match the font in the manuscript. In addition, the error “so can be simplified to the (3) equation as follows” appears in the manuscript, please revise all equations and the manuscript carefully.
3. The font in Figures 3, 9, and 10 is small and inconsistent, please modify them carefully. Further, the quality of all figures should be improved.
4. Simulation conditions are not indicated in this manuscript, please add.
5. “Parasitic motion” and “parasitic displacement” are mentioned several times in the manuscript. What are “parasitic motion” and “parasitic displacement”? Please explain.
6. On rows 307 and 308, the author draws a conclusion that the micromotion platform has good dynamic performance. According to Figure 6, the simulations in different modes were performed, but the author does not compare the simulation results under different modes, so how is the conclusion above reached?
7. Reference 8 is formatted incorrectly, and please check all references.
Please proofread carefully. I suggest seeking assistance from native speakers.
Reviewer 2 Report
The paper studies a STSL amplification
The analysis is complete
Experiments and FE analysis are fully described
There is no real novelty from my point of view but results are well demonstrated
Correct eq.2
Page 8 : i from 1 to 6
Page 10 : what do you mean by "good dynamic performance"?
Page 11 line 4 correct typo
Ref 8 correct typo
Reviewer 3 Report
Dear Authors,
Please revise the manuscript according to attached comments.
Kind Regards

There are some grammatic mistakes. English should be revised as well.
Round 2
Reviewer 3 Report
Dear all,
I do not have any further questions and comments related to the revised version of the manuscript.
Kind Regards
English should be polished.